# Integrated Omics Approach to Discover Differences in the Metabolism of a New Tibetan *Desmodesmus* sp. in Two Types of Sewage Treatments

**DOI:** 10.3390/metabo13030388

**Published:** 2023-03-06

**Authors:** Jinhu Wang, Junyu Chen, Dongdong Zhang, Xiaomei Cui, Jinna Zhou, Jing Li, Yanli Wei, Duo Bu

**Affiliations:** 1College of Ecological Environment, Tibet University, 10 East Zangda Road, Lhasa 850000, China; 2Hunan Eco-Geological Survey and Monitoring Institute, 256 Shaoshan North Road, Yuhua District, Changsha 410000, China

**Keywords:** Tibetan *Desmodesmus* sp., municipal sewage treatment, heavy metal sewage treatment, transcriptomics, metabolomics

## Abstract

Microalgae are now widely applied in municipal (YH_3) and industrial sewage (YH_4) treatments. Through integrated omics analysis, we studied the similarities and differences at the molecular level between the two different types of sewage treatment processes. The most significantly enriched gene ontology (GO) terms in both types of sewage treatments were the ribosome, photosynthesis, and proteasome pathways. The results show that the pathways of differentially expressed genes (DEGs) and differentially accumulated metabolites (DAMs) were enriched for photosynthesis, glyoxylate and dicarboxylate metabolism, and carbon fixation in photosynthetic organisms. Considering YH_3 vs. YH_4, the metabolism of citrate, sedoheptulose-7P, and succinate was significantly upregulated. In addition, the results showed that the pathways of DEGs and DAMs were enriched in terms of amino acid metabolism and carotenoid biosynthesis in YH_4 vs. YH_3. The metabolism of S-Adenosyl-L-homocysteine was significantly downregulated, 2-oxobutanoate was significantly upregulated and downregulated, and the metabolism of abscisic acid glucose ester (ABA-GE) was also significantly upregulated. Overall, the results of this paper will help to improve the basic knowledge of the molecular response of microalgae to sewage treatments, and help design a response strategy based on microalgae for complex, mixed sewage treatments.

## 1. Introduction

For the Asian continent, the Tibetan Plateau plays an important role in environmental and ecological maintenance. Research on the environment and ecology of the Tibetan Plateau is very extensive, and the research on ecological protection deserves continuous attention and funding [1]. Previous studies have shown that the increase in phytoplankton biomass and shifts in community composition in the Tibetan Plateau are mainly affected by global warming [2]. The organic carbon and inorganic carbon soil reserves of grasslands on the Qinghai–Tibet Plateau are also considerable. With the development of climate change, grassland has the potential to become a huge carbon source [3]. In inland aquatic systems, organic matter and dissolved organic carbon play an important role in the carbon cycle [4]. Higher solar radiation, salinity levels, nutrient content, water temperatures, stronger evaporation, and lower pH are all driving mechanisms of gross primary productivity, which affects the protective effects of microalgae on water by promoting or inhibiting their growth and photosynthesis [5].

The advantages of microalgae in domestic and industrial sewage treatments for environmental protection, sustainability, and recycling, are becoming increasingly apparent, with growing applications and research [6]. In addition to recycling, microalgae contain beneficial substances, such as fatty and amino acids post sewage treatment, which can create additional economic value [7,8]. With the acceleration of industrial development and urbanization of residents, the types and components of sewage are becoming increasingly complex [9]. Therefore, increasing attention has been paid to the methods that use recyclable microalgae to treat sewage [10]. In recent years, it has been found that *Desmodesmus* sp. has a good absorption effect on nutrients (carbohydrate, nitrogen, phosphorus) in urban sewage [11], and it is also feasible to carry out the bioremediation of water polluted with excessive levels of heavy metals (copper and nickel) [12]. To better study the mechanisms of microalgae in treating different types of sewage, integrated omics research was employed, and the similarities and differences between microalgae in domestic sewage and industrial sewage treatments were studied at the molecular level [13]. This approach allowed for the verification of the sewage treatment capabilities of microalgae, while also facilitating the study of the molecular mechanisms of microalgae in sewage treatment. Moreover, by studying the metabolic processes in sewage treatment, we can provide theoretical support for extracting the substances with potential economic value from microalgae [14], as well as for the treatment of complex sewage [15].

In previous studies, among the isolated algal strains, *Desmodesmus* sp. was found to have a higher protein yield (34.5 mg L^−1^ day^−1^), accounting for 33.4% of its dry weight, indicating that *Desmodesmus* sp. is a potential algal strain rich in protein feed and nutrition supplement [16]. In a study on the effects of cultivation conditions on *Chlorella* vulgaris and *Desmodesmus* sp. growing in sugarcane biorefinery residues, bagasse and vinasse, we found that the cell density of *Desmodesmus* sp. was higher than that of *Chlorella* vulgaris in terms of metabolite accumulation [17]. The rate of biomass accumulation of *Desmodesmus* sp. under heterotrophic conditions of 40 mM glycerol was similar to that under autotrophic conditions. In addition, under heterotrophic conditions, it produced more fatty acids, all of which are ideal characteristics for biodiesel production [18]. When using microalgae for the bioremediation of excessive nutrients and heavy-metal-contaminated water, *A. mauritanicus* plants and *Desmodesmus* sp. were used to treat single-metal solution wastewater in a sequential approach, and the Cu and Ni removal rates reached up to 74% and 85%, indicating that *Desmodesmus* sp. has a better sewage treatment capacity when used in combination with other organisms [19]. When comparing the removal effects of *Chlorella* vulgaris, *Botryococcus* Braunii and *Desmodesmus* sp. on piggery wastewater, *Desmodesmus* sp. was the best in terms of growth rate, nutrient removal efficiency and content of valuable substances [20]. In water polluted with the heavy metals of copper and nickel, considering the removal ability of two green microalga strains (*Chlorella* vulgaris and *Desmodesmus* sp.) in single-metal solution and mixed heavy metal solution, *Desmodesmus* sp. shows a greater adsorption capacity and higher heavy metal treatment efficiency [21]. Therefore, it is particularly important to further study the Tibetan *Desmodesmus* sp.

The main objectives of this study were as follows: (i) to study the differences in transcriptomics and metabolomics of Tibetan *Desmodesmus* sp. between domestic and industrial heavy-metal sewage treatments; (ii) to find the key metabolic pathways and metabolites of Tibetan *Desmodesmus* sp. in both sewage types, and reveal the differences in the transcription and metabolism of Tibetan *Desmodesmus* sp. between their treatment processes; and (iii) to provide theoretical research on the mechanisms of Tibetan *Desmodesmus* sp. at the molecular level for increasing volumes of sewage with complex components and mixed sources, which is difficult to treat.

## 2. Materials and Methods

### 2.1. Method of Sewage Treatment with Desmodesmus sp. and Sample Preparation

#### 2.1.1. Microalgae and Cultivation

The *Desmodesmus* sp. used in the experiment was isolated from Yamdrok Lake in Tibet, at an altitude of 4300 m. It was identified as a new species by the polyphasic taxonomic method, 18S rRNA sequencing, and the identification and analysis of SNP and InDel loci. It was amplified and cultured in the laboratory for sewage treatment verification [22].

The cultivation and sewage treatment conditions of *Desmodesmus* sp. were a temperature of 25 ± 1 °C, pH 7.0–7.5, light intensity 3000 Lx, and light ratio 12 h:12 h. The BG-11 medium (Hopebio, Qingdao, China) used in the experiment, and containers and consumables in contact with microalgae in contact with microalgae, were sterilized under 101.33 kPa and 121 °C with a steam pressure sterilizer (Shenan, Shanghai, China) for 30 min before use, while breathable sealing film (Bkman, Changde, China) was used to prevent external bacteria from entering. When the cleaning bench (Sujing, Suzhou, China) was open, BG-11 culture solution and simulated municipal and zinc–copper heavy metal sewage were filtered by the vacuum pump (VP-30L, LAB FISH, Huzhou, China) before use, and the water-based membrane filter was 0.45 μm (Xinya, Shanghai, China). Culturing was performed until the logarithmic phase of growth of *Desmodesmus* sp. was achieved and the final optical density OD_689_ value was ≥0.50 (EMCLAB, Duisburg, Germany) [22].

#### 2.1.2. Sewage Treatment Experiment

The zinc–copper heavy metal wastewater was composed of CuSO_4_·5H_2_O (0.037 mg/L) and ZnSO_4_·7H_2_O (0.044 mg/L), simulating the wastewater discharged by an industrial enterprise. The synthetic municipal wastewater was composed of C_6_H_12_O_6_ (375 mg/L), NH_4_Cl (126 mg/L), KH_2_PO_4_ (14 mg/L), NaCl (64 mg/L), FeCl_2_ (4.18 mg/L), NaHCO_3_ (100 mg/L), MgSO_4_ (56.40 mg/L), CaCl_2_ (25 mg/L), H_3_BO_3_ (2.86 mg/L), MnCl_2_·4H_2_O (2.49 mg/L), ZnCl_2_ (0.11 mg/L), Na_2_MoO_4_·2H_2_O (0.39 mg/L), CuCl_2_·2H_2_O (0.054 mg/L), and Co(NO_3_)_2_·6H_2_O (0.049 mg/L). COD: 400.0 ± 0.5 mg/L, TN: 33.0 ± 0.5 mg/L, and TP: 3.2 ± 0.5 mg/L. The pH of wastewater was adjusted to 7.0–7.5 before use [23].

Subsequently, 1 L of the cultured alga solution was centrifuged (Eppendorf, Hamburg, Germany) at 12,000 rpm for 10 min, and the algal mud was washed with BG-11 culture solution three times. The centrifuged algal mud contained in the 1 L of microalgal solution was then added into every 1 L of simulated municipal and zinc–copper heavy metal sewage.

The cultured algal solution was then divided into two parts for three different kinds of sewage treatment. The algal solution before sewage treatment was named YH_2 as the control group, YH_3 as the municipal sewage treatment group, and YH_4 as the zinc–copper heavy metal sewage treatment group. Water samples were collected when the removal rates of chemical oxygen demand, total nitrogen and total phosphorus reached the maximum treatment efficiency for approximately 3–5 days of municipal sewage treatment. Water samples were collected when the removal rates of zinc and copper ions reached the maximum treatment efficiency for 4–6 h of zinc and copper heavy metal sewage treatment. The algal mud, remaining before and after sewage treatment, was washed three times with purified water after centrifugation, and immediately frozen with liquid nitrogen for transcriptomics and metabolomics analyses. Three biological replicates were used for each treatment [24,25].

#### 2.1.3. Detection of Chemical Oxygen Demand, Total Nitrogen, Total Phosphorus, Zinc and Copper

Total nitrogen, total phosphorus and chemical oxygen demand were measured using a water quality multi-parameter tester (Konokeyi KN-MUL20, Beijing, China) and Intelligent digester (Konokeyi KN-HEA12, Beijing, China), following the manufacturer’s instructions. A 0.22 μm stream filter membrane (Xinya, Shanghai, China) was used for filtration prior to the water sample detection. The zinc and copper contents were detected by atomic absorption spectrometer (Agilent 9240FS AA, Santa Clara, CA, USA). According to the instructions of the instrument, the samples were first digested by microwave (Anton Paar 3000, Graz, Austria), and then the zinc and copper contents of the samples were obtained by using the standard curve method [26,27].

### 2.2. RNA-Seq Analysis

#### 2.2.1. RNA Extraction

RNA was isolated from each sample using the Trizol kit (Promega, Cambridge, MA, USA) as per the manufacturer’s instructions. To further understand the transcriptome dynamics of *Desmodesmus* sp. under sewage treatment, the collected pulp tissues were analyzed with RNA-seq with simulated municipal and industrial zinc–copper mixed sewage in three biological replicates. For each sample, RNA-seq libraries were generated using 1 μg RNA, using Illumina’s NEB Next^®^ Ultra^TM^ RNA Library Prep Kit (NEB, Ipswich, MA, USA).

#### 2.2.2. cDNA Synthesis and qPCR

Total RNA was used as the input material for RNA sample preparations. In simple terms, mRNA was purified from total RNA using magnetic beads linked to poly-T oligosaccharides. In the first-strand synthesis reaction buffer (5X), the divalent cations were used for pyrolysis at elevated temperature. The first-strand cDNA was synthesized using a random hexamer primer and M-MuLV Reverse Transcriptase, and the RNA was degraded using RNaseH. The second-strand cDNA was then synthesized using DNA Polymerase I and dNTP. The remaining protrusions were converted to blunt ends by exonuclease/polymerase activity. After adenosine acidification at the 3’ end of the DNA fragments, adapters with a hairpin loop structure were attached to prepare for hybridization. To prioritize the selection of 370–420 bp cDNA fragments, the AMPure XP system (Beckman Coulter, Beverly, CA, USA) was used to purify the library fragments [28,29]. PCR amplification was performed, and PCR products were purified with AMPure XP beads to obtain the library. Libraries were tested to ensure their quality. After the library was constructed, it was quantified using a Qubit 2.0 Fluorometer, then diluted to 1.5 ng/µL, and the library insertion size was determined using an Agilent 2100 bioanalyzer. After the size of the insert reached the desired size, qRT-PCR was used to accurately quantify the effective concentration of the library (i.e., a concentration above 2 nM) to ensure its quality [30].

#### 2.2.3. Illumina Sequencing, Assembly, Annotation and Classification

The library quality was evaluated on an Agilent Bioanalyzer 2100 system, and the libraries were sequenced on the Illumina Novaseq platform by Novagene (Beijing, China). Raw data, in FASTQ format (raw reads), were processed first, and clean reads were obtained by removing reads containing adapters, poly-N, and low-quality reads. 

At the same time, the Q20, Q30, and GC contents of the cleaning data were calculated. The raw sequence reads were deposited in the NCBI (National Center for Biotechnology Information) Sequence Read Archive (SRA) (http://www.ncbi.nlm.nih.gov/subs/sra, accessed on 19 April 2022), accession numbers PRJNA830552 and PRJNA830862). High-quality, clean data were mapped onto the reference genome of *Desmodesmus* sp. using Hisat2 [31].

#### 2.2.4. DEGs and Enrichment Analysis

Gene expression levels were estimated as per kilobase transcriptional sequence per million fragments (FPKM). Differential expression analysis was performed for two groups using the DESeq2 R software package, and *p*-values were adjusted using the Benjamini–Hochberg method to control the error detection rate. Screening of DEGs was carried out according to the following criteria: |log_2_foldchange| > 1 and corrected *p* < 0.05. The selected DEGs were annotated with gene ontology (GO) and the Kyoto Encyclopedia of Genes and Genomes (KEGG).

### 2.3. Metabolite Extraction and Analysis

Microalga samples (100 mg) were ground separately with liquid nitrogen and then resuspended with precooled 80% methanol and 0.1% formic acid in the well vortex. After ice incubation for 5 min, samples were centrifuged at 11,000 rpm and 4 °C for 5 min. Part of the supernatant was diluted with LC-MS-grade water (Thermo Fisher, Waltham, MA, USA) to a final concentration containing 53% methanol. The sample was transferred into a fresh Eppendorf tube and centrifuged at 11,000 rpm and 4 °C for 10 min. Then, the supernatant was injected into the LC-MS/MS system for analysis. UHPLC-MS/MS analysis was performed by the Vanquish UHPLC system (Thermo Fisher, Waltham, MA, USA) and Orbitrap Q Exactive HF mass spectrometer (Thermo Fisher, Waltham, MA, USA) of Novogene Co., Ltd. (Beijing, China). Compound discoverer 3.1 (CD3.1, Thermo Fisher, Hercules, CA, USA) was used to process the raw data file generated by the UHPLC-MS/MS, and the peaks of each metabolite were arranged, extracted, and quantified. Metabolites with variable importance in the projection (VIP) > 1, *p*-value < 0.05, fold change FC > 2 or <0.5 were considered differential metabolites [32]. Principal component analysis (PCA) was carried out using the MetaX [4] software for the logarithmic conversion and centralized formatting of data. The peaks extracted from all of the experimental samples and QC samples were treated by UV, before PCA analysis was carried out. The smaller the difference between QC samples, the better the stability of the whole method and the higher the data quality, which is reflected in the PCA diagram, showing that the distribution of the QC samples gathered together. Scatters of different colors represent samples of different experimental groups, and the ellipses in PCA diagrams have a 95% confidence interval [33].

### 2.4. Combined Transcriptome and Metabolome Analyses

The degree of enrichment of the various pathways was obtained through the co-joint analysis of DEGs and differentially accumulated metabolites (DAMs). The gene–metabolite networks with a Pearson correlation coefficient (PCC) > 0.8 were used to construct the transcript–metabolite network [34].

### 2.5. Statistical Analysis

The statistical software R (R version R-3.6.3, https://www.r-project.org/ (accessed on 10 May 2022)) was used for statistical analysis. When the data were not normally distributed, the area normalization method was tried for normal transformation [35].

## 3. Results

### 3.1. Sewage Treatment Effect and Sample Preparation of Desmodesmus sp.

At a temperature of 25 ± 1 °C, pH 7.0–7.5, light intensity 3000 Lx, and light ratio 12 h:12 h, the highest efficiency of treatment in terms of chemical oxygen demand was 96.1% on the second day, the highest efficiency of treatment in terms of total phosphorus was 90.9% on the fifth day, and the highest efficiency of treatment in terms of total nitrogen was 97.0% on the fifth day. Under the same treatment conditions, the optimal treatment efficiency of zinc and copper was 92.5% and 93.5%, respectively, at the fourth hour. Three biological replicates were sampled at optimum processing times and promptly refrigerated in liquid nitrogen for subsequent analysis after centrifugation and cleaning.

### 3.2. RNA Sequencing and Analysis of DEGs in Non-Parametric Transcriptomics

Transcription and widely targeted metabolite profiles of *Desmodesmus* sp. with both municipal sewage and heavy metal industrial wastewater treatments (YH_3 vs. YH_4) were explored. Three independent biological replicates were used for each treatment, resulting in six samples. The transcriptome changes in *Desmodesmus* sp. were investigated through RNA-Seq analysis. More than 20 million reads were generated per sample. Of these reads, the Q30 percentage (sequencing error rate < 1%) was over 97%, and GC content was ~58% for the libraries. Among all the libraries, 77.72–79.97% of unique reads were mapped to the *Desmodesmus* sp. genome (Table 1). A total of 18,505 genes and 21,511 genes were significantly upregulated and downregulated, respectively, in YH_3 vs. YH_4, (DESeq2 padj < 0.05 |log_2_FoldChange| > 1, Figure 1a,b).

Through gene cluster heatmap analysis, we found that there were significant differences in gene expression between the two comparison pairs (YH_3 vs. YH_4 and YH_4). Therefore, we can conclude that there were significant differences in gene expression between microalgae in the treated municipal and industrial heavy metal sewage (Figure 1c–f).

All of the DEGs were subjected to GO and KEGG analysis. During comparisons of YH_3 and YH_4, we found that the most strikingly enriched GO terms (*p* < 0.01) were linked to biological processes, cellular components, and molecular functions: ribosome and ribonucleoprotein complex biogenesis and translation were the most enriched in the “cellular component” category; ribosome, ribonucleoprotein complex, and intracellular non-membrane-bounded were the most enriched in the “cellular component” category; and structural constituent of ribosome, structural molecule activity, and oxidoreductase activity were the most enriched in the “molecular function” category (Figure 1g, Appendix A).

Pathways showing a significant change (*p* < 0.01) in sewage treatments were identified using the KEGG database. The DEGs were involved in three enriched pathways: ribosome, photosynthesis, and proteasome (Figure 1h, Appendix A).

### 3.3. DAM Analysis 

Principal component analysis (PCA) of the DEGs and differentially accumulated metabolites shows that the YH_3 and YH_4 treatment groups had obvious differences from the YH_2 (control group), which explained 47.52% and 47.56% of the total variation (Figure 2a,b). These results indicate that *Desmodesmus* sp. has a good treatment effect on the two types of sewages. The PCA of the DEGs and DAMs showed clear differences between the YH_3 treatment and YH_4 control groups, which explained 66.29% and 66.69% of the total variation (Figure 2c,d). These results indicate that *Desmodesmus* sp. had a good treatment effect on municipal sewage. Notable differences were also observed between the YH_4 treatment and YH_3 control groups, which explained 65.29% and 66.69% of the total variation (Figure 3a,b). These results indicate that *Desmodesmus* sp. also had a good treatment effect on heavy metal industrial wastewater.

In the scattered plots, the abscissa is the score of the sample on the first principal component, the ordinate is the score of the sample on the second principal component, R2Y represents the interpretation rate of the model, and Q2Y is used to evaluate the prediction ability of the partial least squares discrimination analysis (PLS-DA) model. The PLS-DA models of each comparison group were established, and the model evaluation parameters (R2, Q2) were obtained by a 7-fold cross validation. When the biological repetition number of the sample *n* ≤ 3, it was K cycles of interactive verification (k = 2n); the closer R2 and Q2 were to 1, the more stable and reliable the model was, and when R2Y was greater than Q2Y, the model was well established (Figure 2e,f and Figure 3c,d). The grouping marks of each sample were randomly disrupted before modeling and prediction. Each model corresponded to a set of R2 and Q2 values. Their regression lines could be obtained according to the Q2 and R2 values after 200 disruptions and modeling. When the R2 data were greater than the Q2 data and the intercept between the Q2 regression line and the Y-axis was <0, the model was not “over-fitting”, indicating that the model could better describe the sample and be used as the premise for obtaining the model biomarker group (Figure 2g,h and Figure 3e,f).

For the evaluation of DAMs between YH_3 and YH_4, untargeted metabolomics was applied. After sewage treatment, under the positive polarity mode, 285 metabolites were identified: 160 were significantly upregulated and 125 were significantly downregulated (YH_3 vs. YH_4, VIP > 1.0, FC > 2 or FC < 0.5 and *p*-adjust value < 0.05, Figure 4a, Appendix A). Under the negative polarity mode, 285 metabolites were identified, 160 of which were significantly upregulated and 125 were significantly downregulated (YH_3 vs. YH_4, VIP > 1.0, FC > 2 or FC < 0.5, and *p*-adjust value < 0.05, Figure 4b, Appendix A). 

For the evaluation of DAMs between YH_4 and YH_3, untargeted metabolomics was applied. After sewage treatment, under the positive polarity mode, 547 metabolites were identified, of which 267 were significantly upregulated and 280 were significantly downregulated (YH_4 vs. YH_3, VIP > 1.0, FC > 2 or FC < 0.5, and *p*-adjust value < 0.05, Figure 5a, Appendix A). Under the negative polarity mode, 331 metabolites were identified, of which 144 were significantly upregulated and 187 were significantly downregulated (YH_4 vs. YH_3, VIP > 1.0, FC > 2 or FC < 0.5, and *p*-adjust value < 0.05, Figure 5b, Appendix A).

For the hierarchical cluster analysis (HCA) of DAMs in YH_3 and YH_4 (six replicates for each sample), we found that there were significant differences in the metabolites of *Desmodesmus* sp. between the two different types of sewage (Figure 4c,d and Figure 5c,d). Furthermore, we found that the DAMs were most significantly enriched in the KEGG pathways of zeatin, arginine and alanine biosynthesis, and aspartate and glutamate metabolism (Figure 4e,f); the DAMs in heavy metal sewage treatments were most significantly enriched in the KEGG pathways of arginine, proline, tyrosine, and purine metabolism (Figure 5e,f).

### 3.4. Combined Transcriptome and Metabolome Analyses

To quantitatively map the transcripts directly to the metabolic pathways relevant to *Desmodesmus* sp., comparing YH_3 vs. YH_4, co-joint KEGG pathway enrichment analysis of the transcriptome and metabolome was performed. The results show that the pathways of DEGs and DAMs were enriched for photosynthesis (*p* < 0.01) in the positive polarity mode (Figure 6a), and the same pathways were enriched for glyoxylate and dicarboxylate metabolism and carbon fixation in photosynthetic organisms (*p* < 0.01) in the negative polarity mode (Figure 6b). To better understand the relationship between genes and metabolites, the DEGs and DAMs were simultaneously mapped to the KEGG pathway diagram (Appendix A). As shown in Figure 6c, the metabolites of adenosine diphosphate (ADP) were simultaneously mapped to photosynthesis (ko00195) in the positive polarity mode. In addition, the metabolites of aspartate and sedoheptulose were simultaneously mapped to carbon fixation in photosynthetic organisms (ko00710) in the negative polarity mode (Figure 6d), and the metabolites of citrate, succinate, and L-Serine were simultaneously mapped to the glyoxylate and dicarboxylate metabolism (ko00630) in the negative polarity mode (Figure 6e).

To quantitatively map the transcripts directly to the metabolic pathways involved in *Desmodesmus* sp. and draw comparisons between YH_4 and YH_3, co-joint KEGG pathway enrichment analysis of the transcriptome and metabolome was performed. The results show that the same pathways of DEGs and DAMs were enriched by arginine and proline metabolism (ko00330) and cysteine and methionine metabolism (ko00270) (*p* < 0.01) in the positive polarity mode (Figure 7a), and the same pathways of DEGs and DAMs were enriched by carotenoid biosynthesis (ko00906) (*p* < 0.01) in the negative polarity mode (Figure 7b). To better understand the relationship between genes and metabolites, the DEGs and DAMs were simultaneously mapped on the KEGG pathway diagram (Appendix A). As shown in Figure 7c, the metabolites of ornithine, D-Proline, and 5-amino-pentanoate were simultaneously mapped to arginine and proline metabolism (ko00330) in the positive polarity mode, as shown in Figure 7d, and the metabolites of L-aspartate, L-serine, 2-oxobutanoate, and S-adenosyl-L-methionine were simultaneously mapped to cysteine and methionine metabolism (ko00270) in the positive polarity mode. In addition, the metabolites of abscisate and abscisic acid glucose ester were simultaneously mapped to carotenoid biosynthesis (ko00906) in the negative polarity mode (Figure 7e).

## 4. Discussion

Microalgae are increasingly being used in municipal sewage and industrial heavy metal treatments. The microalga treatment of these types of sewage can avoid secondary pollution and aid recycling, improving the efficiency of sewage treatment, and reducing its cost [36]. At the same time, microalgae utilize different biological mechanisms when treating different types of sewage. The study of microalga metabolism and transcriptomics can provide certain significant guidance for the treatment of mixed sewage, and provide theoretical support for the molecular mechanisms involved [37].

### 4.1. Differences in Expressed Genes and KEGG Enrichment of DEGs

We found that the most strikingly enriched GO terms (*p* < 0.01) were linked to biological processes, cellular components, and molecular function. Pathways showing a significant change (*p* < 0.01) in the two types of sewage treatments were identified using the KEGG database. The DEGs were involved in three enriched pathways: ribosome, photosynthesis, and proteasome. Ribosomes are molecular machines for protein synthesis in cells. The function of the ribosome is to synthesize amino acids into protein polypeptide chains, according to the instructions of the mRNA [38]. Microalga, as a kind of eukaryote, contain protease structures in the nucleus and cytoplasm. The main function of these protease structures is to degrade proteins that are not needed or are damaged. After the degradation of the protease structure, the protein is partitioned into peptides with a length of about approximately 7–8 amino acids. These peptides can be further degraded into single amino acid molecules and then used to synthesize new proteins [39,40]. Photosynthesis can fix carbon, and sugars are synthesized through light cooperation. Therefore, we know that in the two different types of the sewage treatment processes, the synthesis of amino acids, proteins, carbohydrates, and fatty acids is the most significant [41,42,43,44].

### 4.2. Changes in Signaling Pathways and Primary Metabolites

Through the multiple omics analysis of two different types of sewage, we found that the most significant metabolic pathways in municipal sewage treatment were DEGs and DAMs, which were enriched for photosynthesis, glyoxylate, and dicarboxylate metabolism. During the photosynthesis stage of carbon fixation in photosynthetic organisms, the metabolism of ADP is significantly upregulated, which can promote the synthesis of ATP and the water treatment function of microalgae [45]. In glyoxylate and dicarboxylate metabolism, the metabolism of citrate, succinate, and L-serine is significantly upregulated: both citrate and succinate operate in the citrate cycle in microalgae cells and provide a source of energy for wastewater treatment [46,47]. During carbon fixation in photosynthetic organisms, the metabolism of L-aspartate and sedoheptulose-7P is significantly upregulated, which is important to the Calvin–Benson–Bassham (CBB) cycle, the primary metabolic pathway for net CO_2_ fixation within oxygenic photosynthetic organisms [48,49,50]. Compared to heavy metal wastewater treatment, microalgae mainly provide metabolic energy support and amino acid synthesis for municipal wastewater treatment.

The most significant metabolic pathways in heavy metal sewage treatments were the DEGs and DAMs that were enriched for arginine, proline, cysteine, and methionine metabolism and carotenoid biosynthesis. In arginine and proline metabolism, the metabolism of omithine, D-proline, and 5-amino-pentanoate was significantly upregulated, where arginine, omithine, D-proline, and lysine were the most enriched in this process. In cysteine and methionine metabolism, the metabolism of L-serine and L-aspartate was significantly upregulated, and the metabolism of S-adenosyl-L-homocysteine (SAH) was significantly downregulated. SAH is an amino acid derivative and a key intermediate metabolite in methionine metabolism, which is normally considered a harmful by-product that hydrolyzes quickly once formed. Microalgae are able to control and reduce levels of this harmful metabolite to maintain normal metabolism and better treat sewage [51]. The metabolism of 2-oxobutanoate (2-OBA) was both significantly upregulated and downregulated. When microalgae absorb nutrients to synthesize amino acids, carbohydrates, and fatty acids in the process of sewage treatment, they mainly focus on primary metabolism, and the metabolite 2-OBA is significantly downregulated. When the nutrients in sewage are consumed, microalgae produce a secondary metabolic process, which leads to the significant upregulation of 2-OBA. As a toxic metabolic intermediate, 2-OBA generally arrests the cell growth of most microorganisms and blocks the biosynthesis of target metabolites [52,53]. It is a secondary metabolite of propanoate metabolism. The production of the oxidative stress marker ophthalmate, and the branched-chain keto acids 2-OBA, as a metabolite of branched-chain amino acid aminotransferases (BCATs), catalyzes reversible transamination between them [54,55]. When wastewater is deficient in nutrients, amino acid metabolism in microalgae induces 2-OBA under the action of amino acid transaminase, so as to induce the secondary metabolic process of propanoate metabolism and the oxidative stress marker ophthalmate [56]. Therefore, in order to obtain fatty acids, amino acids, and other substances of economic value in microalgae, it is necessary to harvest them when the clearance rate is best, and then use them for subsequent treatment, which can reduce the occurrence of secondary metabolic processes. In carotenoid biosynthesis, the metabolism of abscisic acid glucose ester (ABA-GE) was significantly upregulated, and the metabolism of abscisate (ABA) was both significantly upregulated and downregulated. In the treatment of heavy metal wastewater, ABA-GE is catalyzed by b-glucosidase to form ABA, counteracting the effects of heavy metal salt stress. The biosynthesis of ABA-GE, amino acids, and 2-OBA are the most significant metabolic paths in microalgae when used to treat heavy metal wastewater under the stress of heavy metals, compared with processing municipal sewage [57,58].

Through discussion in this study, we found that citrate and succinate provided energy sources for wastewater treatment. The harmful metabolism of S-adenosyl-L-homocysteine was downregulated, 2-oxobutanoate induced the secondary metabolic procession, and the deconjugation of abscisic acid glucose ester aided in the elimination of heavy metal stress.

## 5. Conclusions

Tibetan *Desmodesmus* sp. has good application prospects in municipal and heavy metal sewage treatments. In the process of comparatively studying the mechanisms of sewage treatment, we found that the metabolites of amino acids were most enriched in sewage treatment.

In addition, we found that the response of *Desmodesmus* sp. to municipal sewage treatment could significantly enhance carbon fixation in photosynthetic organisms. We also found that its response to heavy metal sewage treatment could induce the conversion between ABA and ABA-GE, facilitate the formation of free ABA, and in the absence of nutrients, also induce the secondary metabolism of propanoate and the oxidative stress marker ophthalmate, in order to offset the environmental impact. This provides theoretical support for our understanding of microalgae adapting to heavy metal stress.

Overall, microalgae can cause significant changes in amino acid metabolism during municipal wastewater treatment. While microalgae tend to metabolize amino acids in the treatment of industrial heavy metal wastewater, they will upregulate corresponding metabolites in response to heavy metal stress to maintain normal metabolism. Through the integrated omics study of two different kinds of sewages, we can further understand the mechanisms of sewage treatment, and provide some reference data for the microalgal treatment of complex, mixed sewage. The study of the relevant mechanisms in the wastewater treatment process of Tibetan *Desmodesmus* sp. provides theoretical support for its use in the recycling of medicine and biomass energy in the future. In addition, the study of other microalgae in Tibet needs follow-up and continuous research.

## Figures and Tables

**Figure 1 metabolites-13-00388-f001:**
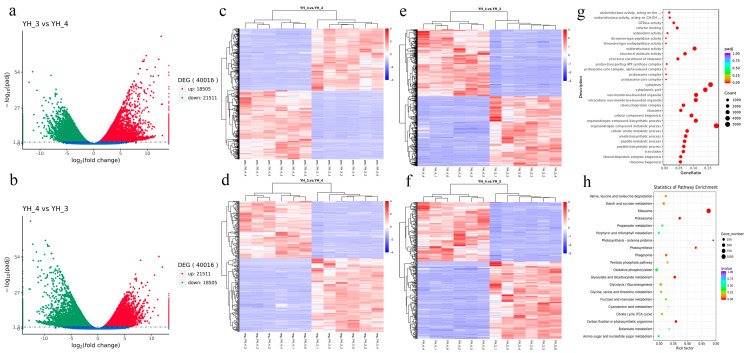
Functional annotations of the number of DEGs, GO classification, cluster heatmap of differentially expressed genes, and KEGG enrichment of DEGs. (**a**) Number of DEGs with YH_3 vs. YH_4. (**b**) Number of DEGs with YH_4 vs. YH_3. (**c**) Cluster heatmap of differentially expressed genes in positive polarity mode. (**d**) Cluster heatmap of differentially expressed genes in negative polarity mode. (**e**) Cluster heatmap of DEGs in positive polarity mode. (**f**) Cluster heatmap of DEGs in negative polarity mode. The abscissa in the figure represents the clustering results of samples, and the ordinate represents the clustering results of differential genes. The color represents the expression level of genes in the sample. The redder, the higher the expression, and the greener, the lower the expression. (**g**) GO classification. (**h**) KEGG enrichment of DEGs.

**Figure 2 metabolites-13-00388-f002:**
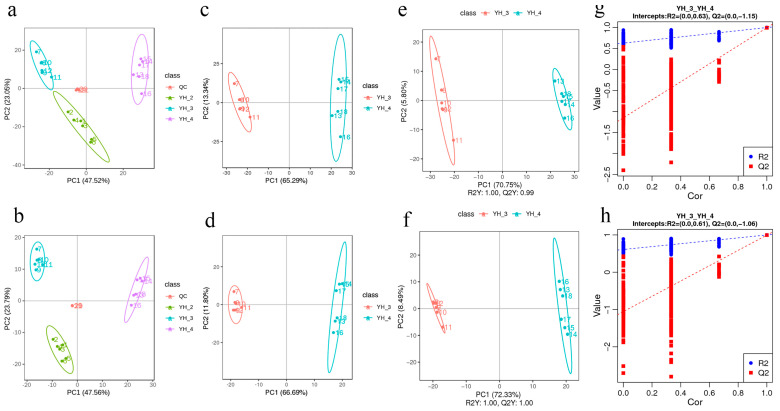
PCA in YH_2 vs. YH_3 vs. YH_4. (**a**) PCA of the variance-stabilized estimated raw counts of DAMs in positive polarity mode in YH_2 vs. YH_3 vs. YH_4. (**b**) PCA of the variance-stabilized estimated raw counts of DAMs in negative polarity mode in YH_2 vs. YH_3 vs. YH_4. (**c**) PCA of the variance-stabilized estimated raw counts of DAMs in positive polarity mode in YH_3 vs. YH_4. (**d**) PCA of the variance-stabilized estimated raw counts of DAMs in negative polarity mode in YH_3 vs. YH_4. (**e**) PLS-DA of DAMs in positive polarity mode. (**f**) PLS-DA of DAMs in negative polarity mode in YH_3 vs. YH_4. (**g**) Permutation test for positive polarity mode. (**h**) Permutation test for negative polarity mode in YH_3 vs. YH_4.

**Figure 3 metabolites-13-00388-f003:**
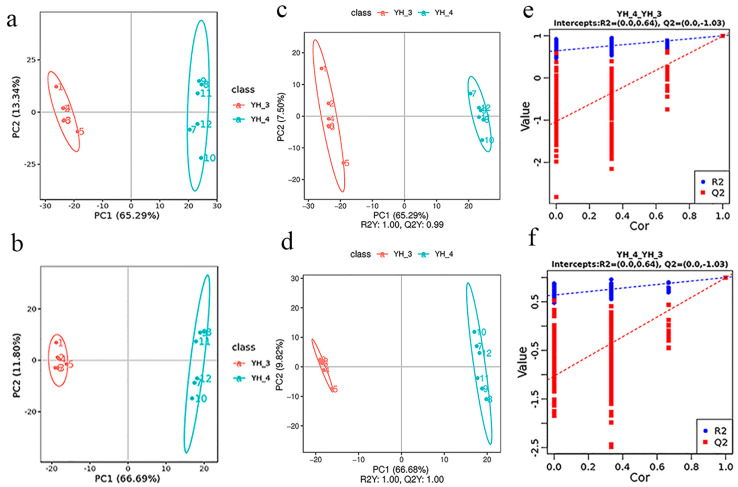
PCA in YH_4 vs. YH_3. (**a**) PCA of the variance-stabilized estimated raw counts of DAMs in positive polarity mode in YH_3 vs. YH_4. (**b**) PCA of the variance-stabilized estimated raw counts of DAMs in negative polarity mode in YH_3 vs. YH_4. (**c**) PLS-DA of DAMs in positive polarity mode in YH_3 vs. YH_4. (**d**) PLS-DA of DAMs in negative polarity mode in YH_3 vs. YH_4. (**e**) Permutation test for positive polarity mode in YH_3 vs. YH_4. (**f**) Permutation test for negative polarity mode in YH_3 vs. YH_4.

**Figure 4 metabolites-13-00388-f004:**
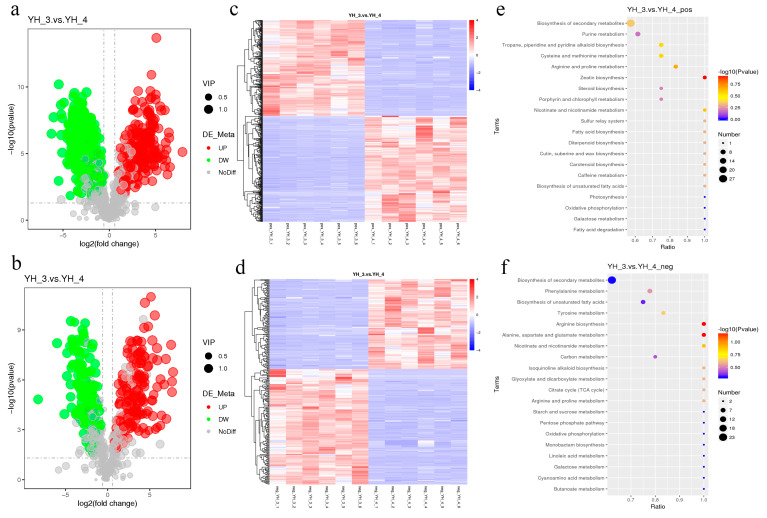
DAMs analysis in YH_3 vs. YH_4. (**a**) Volcano plot of differential metabolites in positive polarity mode. (**b**) Volcano plot of differential metabolites in negative polarity mode. Each point in the volcano map represents a metabolite, and the X-axis represents the logarithm of the quantitative difference of a certain metabolite in the two samples; the Y-axis represents the significance level of the difference (−log10p value). The green dots represent downregulated DAMs, the red dots represent upregulated DAMs, and the gray dots represent DAMs that were not significantly changed. (**c**) HCA of DAMs in YH_3 vs. YH_4 in positive polarity mode. (**d**) HCA of DAMs in YH_3 vs. YH_4 in negative polarity mode. (**e**) Scatter plot of KEGG pathways enriched in YH_3 vs. YH_4 in positive polarity mode. (**f**) Scatter plot of KEGG pathways enriched in YH_3 vs. YH_4 in negative polarity mode. The abscissa in the figure represents the ratio of the number of differential metabolites in the corresponding metabolic pathway, to the number of total metabolites identified in the pathway. The greater the value, the higher the degree of enrichment of differential metabolites in the pathway. The color of the point represents the *p*-value value of the hypergeometric test. The smaller the value, the greater the reliability and statistical significance of the test. The size of the point represents the number of differential metabolites in the corresponding pathway. The larger the point, the more differential metabolites in the pathway.

**Figure 5 metabolites-13-00388-f005:**
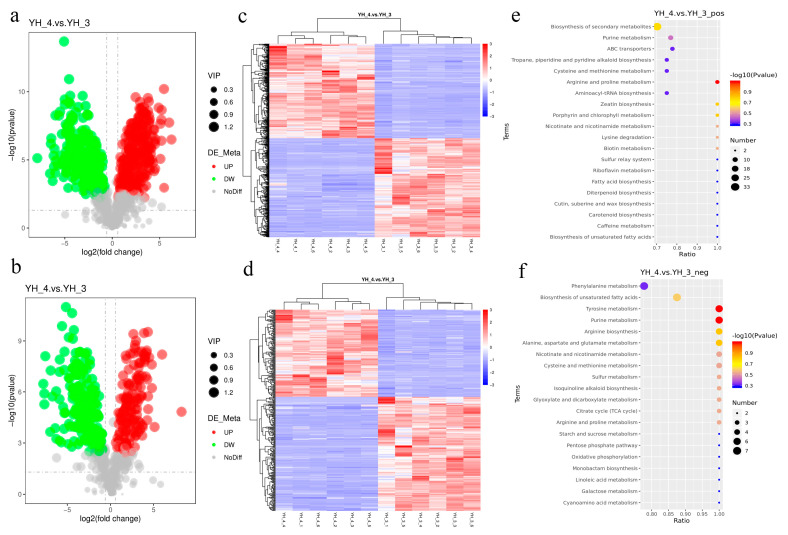
DAMs analysis in YH_4 vs. YH_3. (**a**) Volcano plot of differential metabolites in positive polarity mode. (**b**) Volcano plot of differential metabolites in negative polarity mode. Each point in the volcano map represents a metabolite, and the X-axis represents the logarithm of the quantitative difference of a certain metabolite in the two samples; the Y-axis represents the difference in the significance level (−log10p value). The green dots represent downregulated DAMs, the red dots represent upregulated DAMs, and the gray dots represents DAMs that were detected but not significantly changed. (**c**) HCA of DAMs in YH_4 vs. YH_3 in positive polarity mode. (**d**) HCA of DAMs in YH_4 vs. YH_3 in negative polarity mode. (**e**) Scatter plot of KEGG pathways enriched in YH_4 vs. YH_3 in positive polarity mode. (**f**) Scatter plot of KEGG pathways enriched in YH_4 vs. YH_3 in negative polarity mode. The abscissa in the figure represents the ratio of the number of differential metabolites in the corresponding metabolic pathway, to the number of total metabolites identified in the pathway. The greater the value, the higher the degree of enrichment of differential metabolites in the pathway. The color of the point represents the *p*-value of the hypergeometric test. The smaller the value, the greater the reliability and statistical significance of the test. The size of the point represents the number of differential metabolites in the corresponding pathway. The larger the point, the more differential metabolites in the pathway.

**Figure 6 metabolites-13-00388-f006:**
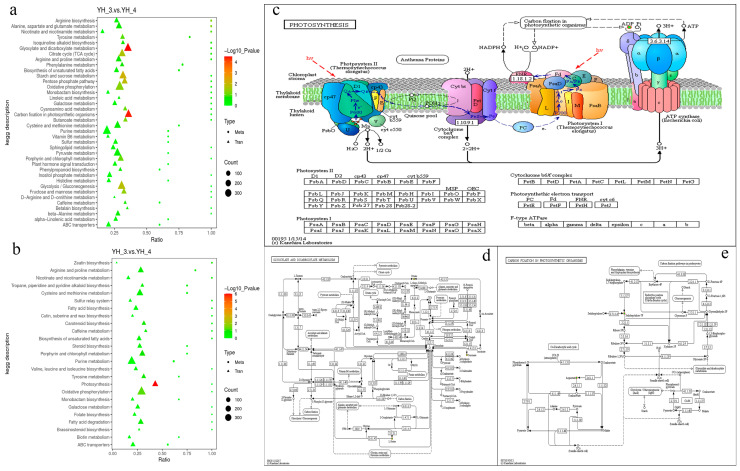
Joint analysis of the DEGs and DAMs in YH_3 vs. YH_4. (**a**) Scatter plot of differential genes and metabolites in positive polarity mode. (**b**) Scatter plot of differential genes and metabolites in negative polarity mode. (**c**) The DEGs and DAMs simultaneously mapped to photosynthesis (ko00195). (**d**) The DEGs and DAMs simultaneously mapped to glyoxylate and dicarboxylate metabolism (ko00630). (**e**) The DEGs and DAMs simultaneously mapped to carbon fixation in photosynthetic organisms (ko00710). Solid circles in green indicate the metabolites noted, red circles represent a significantly upregulated gene or metabolite, blue circles represent a significantly downregulated gene or metabolite, and solid circles in green represent a gene that is both upregulated and downregulated.

**Figure 7 metabolites-13-00388-f007:**
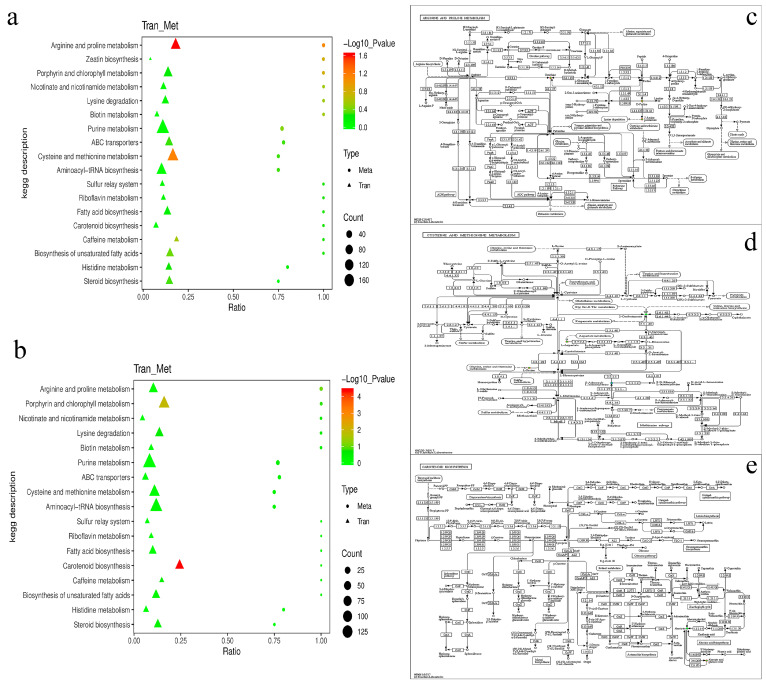
Joint analysis of the DEGs and DAMs in YH_4 vs. YH_3. (**a**) Scatter plot of differential genes and metabolites in positive polarity mode. (**b**) Scatter plot of differential genes and metabolites in negative polarity mode. (**c**) The DEGs and DAMs simultaneously mapped to arginine and proline metabolism (ko00330). (**d**) The DEGs and DAMs simultaneously mapped to the cysteine and methionine metabolism (ko00270). (**e**) The DEGs and DAMs simultaneously mapped to carotenoid biosynthesis (ko00906). Solid circles in green indicate the metabolites noted, red circles represent a significantly upregulated gene or metabolite, blue circles represent a significantly downregulated gene or metabolite, and solid circles in green represent a gene that is both upregulated and downregulated.

**Table 1 metabolites-13-00388-t001:** Sample sequencing data quality summary and comparison statistics. YH_3_1–YH_3_3 refer to three replicates with municipal sewage; YH_4_1–YH_4_3 refer to three replicates with industrial heavy metal wastewater.

Sample	Raw_Reads	Clean_Reads	Clean_Bases	Error_Rate	Q20	Q30	GC	Total Map
YH_3_1	21,721,828	20,683,812	6.2 G	0.02	98.14%	94.49%	58.56%	32,435,152 (78.41%)
YH_3_2	23,353,672	21,917,678	6.6 G	0.03	97.20%	92.20%	58.51%	34,397,556 (78.47%)
YH_3_3	21,446,868	20,184,798	6.1 G	0.02	98.14%	94.49%	59.27%	31,376,358 (77.72%)
YH_4_1	23,642,040	22,206,404	6.7 G	0.02	98.12%	94.50%	58.48%	35,514,902 (79.97%)
YH_4_2	21,109,910	19,405,857	5.8 G	0.02	98.34%	94.87%	57.97%	30,997,286 (79.87%)
YH_4_3	22,696,797	21,629,448	6.5 G	0.02	98.10%	94.45%	57.96%	33,996,226 (78.59%)

Q20 refers to the percentage of bases with a Phred value greater than 20, and Q30 refers to the percentage of bases with a Phred value greater than 30 where Phred = −10log10(e).

## Data Availability

The raw sequence reads were deposited in the NCBI Sequence Read Archive (http://www.ncbi.nlm.nih.gov/subs/sra (accessed on 19 April 2022), accession number PRJNA830552; accession number PRJNA830862).

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
