# Peer review of "Integrated Omics Approach to Discover Differences in the Metabolism of a New Tibetan Desmodesmus sp. in Two Types of Sewage Treatments"

_metabolites, 2023, doi:10.3390/metabo13030388_

Round 1

Reviewer 1 Report

The article is interesting but some explanations need to be added for improving clarity. Moderate revision.

Authors should explain more:

1. Please add an introduction comparison of Desmodesmus sp with other kinds of species like Haematococcus pluvialis or Chlorella vulgaris. Add advantages of commercially potential byproducts occurring at the approach of Desmodesmus. And add reason of choice of Desmodesmus sp.

2. Please in line 97 correct formula put subscript as the number of elements

3. Why You did not apply Bonferroni correction to error regression?

Author Response

Dear Reviewer #1,

Thank you for your comments. These comments are all valuable and very helpful for revising and improving our paper, as well as the important guiding significance to our researches. We read the comments carefully and made some correction. According to the review suggestions, we have supplied the necessary information in the introduction, methods, results, discussion and conclusion parts. The main corrections and the main responds to the reviewer’s comments are submitted in this paper.

Reply to Reviewer #1

Thank you very much for giving us an opportunity to revise our manuscript, and we also appreciate you very much for your positive and constructive comments and suggestions on our manuscript.

Comments: “The article is interesting but some explanations need to be added for improving clarity. Moderate revision.

Authors should explain more:

  1. Please add an introduction comparison of Desmodesmussp with other kinds of species like Haematococcuspluvialis or Chlorella vulgaris. Add advantages of commercially potential byproducts occurring at the approach of Desmodesmus. And add reason of choice of Desmodesmus sp.
  2. Please in line 97 correct formula put subscript as the number of elements.
  3. Why You did not apply Bonferroni correction to error regression?” 

Point 1: “Please add an introduction comparison of Desmodesmus sp with other kinds of species like Haematococcus pluvialis or Chlorella vulgaris. Add advantages of commercially potential byproducts occurring at the approach of Desmodesmus. And add reason of choice of Desmodesmus sp.”

Response 1: We appreciate your clear and detailed feedback and hope that the explanation has fully addressed all of your concerns. Your comments made us aware of the problem with the manuscript. Therefore, fully consider your comments and make revisions in the manuscript. 

First of all, we modified and corrected the introduction, methods and conclusion section, strengthened the research background of the manuscript, and checked the relevance of the references, modified the content of the introduction, and improved the expression of the relevant content according to the content of the references. References have been added to further strengthen the introduction section in line 67-88. 

In the methods section part, some experimental details are added, and the method part is organized to make the experimental method and design process clearer.

The results section has been added with the introduction of the experimental results in line 237-242, and we also improved the small lettering in the figures, so that readers can understand the author's intention when reading the results section. Your comments are very important to the revision of the manuscript, and we look forward to your comments.

Point 2: “Please in line 97 correct formula put subscript as the number of elements.”

Response 2: Thank you very much for your comments, your comments are very professional and detailed. We made some changes to the methods part in line 119-125.

Point 3: “Why you did not apply Bonferroni correction to error regression?”

Response 3: Thank you very much for your professional review. P-value algorithm is to perform log2 conversion on the relative quantitative value in the meta intensity table when we are screening differential metabolites. After this step is completed, the data has been approximated to the normal distribution, and the two-tailed test in T-test can be used to calculate the p value. Generally, adjust P value is a more strict P value, which is generally used for genomics with large data volume such as gene or transcriptome; while the pvalue value is generally used for genomics with small data volume such as protein or metabolism. In this study, Tibetan Desmodesmus sp. belong to species with small genome, so P Value is adopted in practical application. Thank you again for your professional reminder, we will focus on this point in the future study of some species with large genome.

We appreciate your clear and detailed feedback and hope that the explanation has fully addressed all of your concerns. Thank you very much for your professional review and pointing out the direction for our manuscript. We sincerely hope that this manuscript can be published and look forward to your review.We would like to take this opportunity to thank you for all your time involved and this great opportunity for us to improve the manuscript. We hope you will find this revised version satisfactory.

I wish this revision will be acceptable for publication in Metabolites.

Thank you for your consideration. I am looking forward to hearing from you.

Yours Sincerely,

Wang Jinhu

Address: Lhasa, Tibet, China.

Email: phudor@vip.163.com

Tel: +8613618465558

Reviewer 2 Report

In the paper entitled: Integrated omics approach to discover differences in the metabolism of a new Tibetan Desmodesmus sp. in two types of sew-age treatment  the authors through integrated omics analysis studied the similarities and differences at the molecular level between the two different types of sewage treatment processes.

This study is original and useful. Topic is very interesting enough to attract the readers’ attention.  The study is conducted with scientific rigor and meticulousness. This work could be published after minor revision. 

My observations are as follows: 

·       At lines 97-98 check the subscripts. There are some missing 

·       I would improve the figures because in some the lettering is too small and you can't read it 

·       I would also make a very simple final outline illustrating the content of their findings

Author Response

Response to Reviewer #2 Comments

Dear Reviewer #2,

Thank you for your letter and comments. These comments are all valuable and very helpful for revising and improving our paper, as well as the important guiding significance to our researches. We read the comments carefully and made some correction. According to the review suggestions, we have supplied the necessary information in the methods, results and disscusion part. The main corrections and the main responds to the reviewer’s comments are submitted in this paper.

Reply to Reviewer #2

Thank you very much for giving us an opportunity to revise our manuscript, and we also appreciate you very much for your positive and constructive comments and suggestions on our manuscript.

Comments: “In the paper entitled: “Integrated omics approach to discover differences in the metabolism of a new Tibetan Desmodesmus sp. in two types of sewage treatment”  the authors through integrated omics analysis studied the similarities and differences at the molecular level between the two different types of sewage treatment processes.

This study is original and useful. Topic is very interesting enough to attract the readers’ attention.  The study is conducted with scientific rigor and meticulousness. This work could be published after minor revision. 

My observations are as follows: 

      At lines 97-98 check the subscripts. There are some missing. 

       I would improve the figures because in some the lettering is too small and you can't read it. 

      I would also make a very simple final outline illustrating the content of their findings.”

Point 1: “At lines 97-98 check the subscripts. There are some missing.”

Response 1: Thank you very much for your comments, your comments are very professional and detailed. We made some changes to the methods part in line 119-125.

Point 2: “I would improve the figures because in some the lettering is too small and you can't read it.”

Response 2: Thank you very much for your professional review, which was a great help in revising the manuscript. It is really true as Reviewer suggested that lettering from picture 1 to 7 in the manuscript are very small. Therefore, we have made appropriate adjustments to the letters in the manuscript. We are very sorry for our negligence.

Point 3: “I would also make a very simple final outline illustrating the content of their findings?”

Response 3: We appreciate your clear and detailed feedback and hope that the explanation has fully addressed all of your concerns. Your comments made us aware of the problem with the manuscript. Therefore, fully consider your comments and make revisions in the manuscript. First of all, we modified and corrected the discussion section, we made a very simple final outline illustrating the content of our findings in line 528-532. Thank you very much for your professional review, which was a great help in revising the manuscript.

We appreciate your clear and detailed feedback and hope that the explanation has fully addressed all of your concerns. Thank you very much for your professional review and pointing out the direction for our manuscript. We sincerely hope that this manuscript can be published and look forward to your review.We would like to take this opportunity to thank you for all your time involved and this great opportunity for us to improve the manuscript. We hope you will find this revised version satisfactory.

I wish this revision will be acceptable for publication in Metabolites.

Thank you for your consideration. I am looking forward to hearing from you.

Yours Sincerely,

Wang Jinhu

Address: Lhasa, Tibet, China.

Email: phudor@vip.163.com

Tel: +8613618465558

Reviewer 3 Report

Authors presented an interesting work reporting application of microalage for the sewage treatment, authors performed lot of measurements and presented their finding , I would appreciate inclusion of Design of Experiments (DoE) in their work to clearly show the effect of different parameters on the efficiency of the treatment procedure, rest looks fine but DoE is must before publication

Author Response

Response to Reviewer #3 Comments

Dear Reviewer #3,

Thank you for your letter and comments. These comments are all valuable and very helpful for revising and improving our paper, as well as the important guiding significance to our researches. We read the comments carefully and made some correction. According to the review suggestions, we have supplied the necessary information in the introduction, methods, results, discussion and conclusion parts. The main corrections and the main responds to the reviewer’s comments are submitted in this paper.

Reply to Reviewer #3

Thank you very much for giving us an opportunity to revise our manuscript, and we also appreciate you very much for your positive and constructive comments and suggestions on our manuscript.

Comments 1: “Authors presented an interesting work reporting application of microalage for the sewage treatment, authors performed lot of measurements and presented their finding, I would appreciate inclusion of Design of Experiments (DoE) in their work to clearly show the effect of different parameters on the efficiency of the treatment procedure, rest looks fine but DoE is must before publication”

Response 1: First of all, we modified and corrected the introduction, methods and conclusion section, strengthened the research background of the manuscript, and checked the relevance of the references, modified the content of the introduction, and improved the expression of the relevant content according to the content of the references. References have been added to further strengthen the introduction section in line 67-88.

In the methods section, some experimental details are added, and the method part was reorganized to make the experimental method and design process clearer.

The results section has been added with the introduction of the experimental results in line 236-242, and we also improved the small lettering in the figures, so that readers can understand the author's intention when reading the results section. Your comments are very important to the revision of the manuscript, and we look forward to your comments.

Comments 2: “Extensive editing of English language and style required.”

Response 2: Thank you for your careful observation to make the revision of the manuscript more specific. We have carefully and thoroughly proofread the manuscript to correct all the grammar and typos. MDPI for its linguistic assistance during the preparation of this manuscript.

English-Editing-Certificate-61585

We appreciate your clear and detailed feedback and hope that the explanation has fully addressed all of your concerns. Thank you very much for your professional review and pointing out the direction for our manuscript. We sincerely hope that this manuscript can be published and look forward to your review.We would like to take this opportunity to thank you for all your time involved and this great opportunity for us to improve the manuscript. We hope you will find this revised version satisfactory.

I wish this revision will be acceptable for publication in Metabolites.

Thank you for your consideration. I am looking forward to hearing from you.

Yours Sincerely,

Wang Jinhu

Address: Lhasa, Tibet, China.

Email: phudor@vip.163.com

Tel: +8613618465558

Round 2

Reviewer 1 Report

Accepted. Thank You very much for the wide and complete answers to my questions.

Reviewer 3 Report

Ok paper can be accepted, but i would urge authors should spend some time what  is ment by DoE (design of experiments)